# Maternal tobacco smoking and childhood obesity in South Africa: A cohort study

**Nomusa Y. Nkomo, Beatrice D. Simo-Kengne** **\*, Mduduzi Biyase**

School of Economics, College of Business and Economics, University of Johannesburg, Johannesburg, South Africa

\* bdsimo-kengne@uj.ac.za

## Abstract

### Background

Tobacco use and obesity are major public health problems and part of the leading causes of preventable disease and death worldwide. The prevalence of overweight children has escalated over the years; making the weight gain in children a critical issue for health professionals and academics alike. This study examines the association between maternal tobacco use and body weight abnormalities among South African children.

### Data and methods

The study uses data from the National Income Dynamics Study (NIDS), conducted from 2008 to 2017. The survey is available in five waves which have been merged into a panel dataset of 211,718 children aged between 0 to 5 years old, drawn from a nationally representative sample of households followed across time. Thus, the subgroup of children aged between 6 and 15 years old was excluded from the analysis. The empirical investigation employs a logistic regression model to estimate the marginal effect of maternal smoking on childhood overweight prevalence measured by three anthropometric based proxies, namely risk of child overweight, child overweight and child obesity. This framework assumes unobserved child characteristics to be uncorrelated with independent variables (random effect assumption); hence allowing to control for time-invariant sociodemographic factors which are likely to affect child nutritional health.

### Results

In addition to sociodemographic and health characteristics of mothers, empirical findings suggest that maternal smoking exhibits a significant odds and/ or probability of weight abnormalities in children. Notably, children of smoking mothers are likely to be obese, overweight, or at risk of becoming overweight with a possible coexistence of mother-child overweight. Similarly, weight irregularities in children are likely to increase with mother's age. Conversely, married mothers although associated with increased odds of children having high BMI, display a reduced probability of children being abnormally weighted. Finally, child support grant has the potential to improve children nutritional health as children whose mothers are recipient of such grant have a negative probability of having high BMI.

**Data Availability Statement:** Data are available from the link: http://www.nids.uct.ac.za/nids-data/data-access.

**Funding:** The authors received no specific funding for this work.

**Competing interests:** he authors have declared that no competing interests exist.

## Conclusion

Overall, maternal smoking contributes to child overweight and/or obesity in south Africa; suggesting that maternal healthy lifestyle could be an alternative strategic tool to fight against overweight in children. However, mothers' inability to remain and/or follow a healthy life style is plausible as age increases, with spillover effect on child care. Thus, policymakers should prioritize programs to reduce smoking, especially amongst pregnant women and caregivers, to minimise the risk of overweight in children. Promoting the consumption of healthy foods accompanied by physical activity may reduce mothers' stress levels and their incentive to self-medicate using tobacco substances. In addition, comprehensive action programs including child support grant and recommendations for treatment plans that address the problems of children who are already suffering from overweight, remain essential.

## 1. Introduction

There seems to be agreement that the prevalence of overweight in children is not explained only by genetic transfer, but also by surrounding environmental factors and lifestyle choices. According to the Life History Theory, exposure to random environments during infancy contributes to an individual being overweight in adulthood [1]. Childhood environments and unhealthy diet result in overweight challenges during maturity. For instance, uncertain and scarce resources create environments that encourage mothers to overfeed themselves and their children out of fear of not having food in future, thus causing the overweight transfer from mother to child.

Overweight may be caused by non-genetic factors driven by historical, social and cultural behaviors that have steered the control of fat cells [2]. Accordingly, as most mothers become physically inactive during pregnancy, the development muscles dominated by fat cells due to their compromised metabolism is transferred to their unborn babies, resulting in increased overweight in infants. Consistent with the Early Life Theory, weight problems may start early in a child's life due to social and cultural evolution factors.

This overweight problem is a significant burden on families and healthcare systems. Overweight among women of childbearing age has increased [3] and some studies have shown that parents who are overweight also contribute to overweight in their children and future overweight adults [4]. While Children have progressively shifted to indoor entertainment such as television and computer games [5] resulting in minimal physical activities, the incidence of overweight is generally prevalent in disadvantaged communities, resulting from being exposed to lifestyle influences such as an unhealthy diet, cigarette smoking, and lack of exercise.

Another environmental driver of overweight in children may be parental lifestyle choices. Parental home smoking has been seen as an important predictor of overweight in childhood [4]. Of particular interest is maternal smoking, which is considered to be a highly modifiable risk factor linked to children's obesity [4]. Regular smokers of tobacco substances, including current and future mothers, tend to accumulate fats [5] and children of overweight mothers have a higher risk to have abnormal weight at birth or at a later stage [6]. This may happen as a result of their genes or of breastfeeding [7]. Although less likely to breastfeed, smoking mothers tend to be overweight, are generally of low socioeconomic status with less education [8].

Overweight in children under five years of age is common in low- and middle-income countries, including South Africa. In the African region, childhood overweight has doubled

from 1990 to 2013 [14]; the 2017 obesity prevalence among children under the age of five reaching 10.3 percent, and 13.7 percent in North Africa and South Africa, respectively [5]. This overweight condition seems to prevail in children whose parents are overweight and children exposed to maternal smoking before and during pregnancy [7]. In developing countries, the problem of overweight seems to be more prevalent in rich communities in urban areas, as they adapt to the western lifestyle [9]. Marketing unhealthy food and higher levels of stress from studying are some of the important drivers of child overweight; females being significantly more overweight than males [9]. In addition, tradition and culture have also contributed a fair share to overweight, as they believe that fat (overweight) children are healthy and that the weight will disappear as the child grows up [5].

However, empirical findings on the association between maternal smoking and childhood overweight remain controversial, with the underlying mechanism being obscure given the possible overlap between genetic and environmental factors, including diet [10] and behavioural habits [11]. Depending on the context, maternal smoking may even be linked to child weight loss rather than child weight gain [12], or have no weight effect at all [13]. Considering the high level of smoking and overweight challenges in South Africa [6], this study hypothesizes and tests the overweight children's likelihood to have regular smoking mothers. Existing literature on the implications of maternal smoking for their children's health in South Africa mainly focuses on maternal smoking during pregnancy [6, 9]. This paper extends the previous literature by providing a longitudinal analysis of mothers' smoking status and its impact on their offspring's weight from infancy to early childhood.

While smoking is believed to comfort or relieve people from the pressures of life, Buckley [14] has indicated that smoking increases their rate of anxiety and tension. This particular feeling may result in an increased prevalence of overweight and other non-communicable diseases as individuals turn to food to self-medicate their feelings. At least 17.6% of the South African population smokes. The smoking rate of women aged 18 and above has increased to 7.3% [15]. Similar trend is observed with weight problems; almost 70% of females and 40% of males are either overweight or obese, two-thirds of adult women being overweight in South Africa, which indicates that the childhood overweight problem will worsen as these children grow older [16]. This is a significant health risk because women are present and future mothers, and an increasing number of smoking women causes a larger number of exposed children. The rate of overweight in South Africa is rapidly increasing with 14.2% of overweigh children recorded in primary grades [17]. It is assumed that if this problem grows at the current rate, 3.91 million children will be obese by 2025 [18]. Therefore, preventing childhood overweight is not only about improving children's health, although that is an important goal, but is also linked to avoiding obesity and untimely deaths among future adults. People with obesity are more likely to experience cardiovascular disease (CVD), cancer and type 2 diabetes and have an increased mortality risk compared to their normal-weight partners [19]. Accordingly, understanding the contributing factors to childhood obesity becomes imperative in addressing overweight-related future public health challenges in South Africa. Building on existing literature, this study investigates the possible association between maternal smoking and child weight abnormalities in south Africa using a panel logistic regression model.

## 2. Materials and methods

### 2.1 Data

This study uses the National Income Dynamics Study (NIDS) survey, a longitudinal household study in South Africa conducted by the Southern Africa Labour and Development Research Unit (SALDRU) based at the University of Cape Town's School of Economics. Started in 2008,

NIDS is repeated every two years by tracking a nationally representative sample of the same households across the country to collect information on people livelihood using (i) the households questionnaire (on household characteristics, household roster, mortality history, living standards, expenditure, consumption, negative events, positive events, agriculture), (ii) the adults questionnaire (on demographics, education, labour market participation, income, health, well-being, numeracy, anthropometric data) and (iii) the children questionnaire (on education, health, family support, grants, anthropometric data, numeracy).

To investigate the association between maternal smoking and child health, a cohort study is performed on children aged between 0 to 5 years old over the period from 2008 to 2017. Since children are defined in South Africa as persons below the age of 15, the subgroup of children aged between 6 and 15 years old was excluded from the analysis. The survey is available in five waves which have been merged into a panel dataset of 62 243 children arranged according to child person identifier. Three different child nutritional health indicators are used as dependent variable, namely child risk of being overweight, child overweight, and child obesity. These child health variables were created using the Body Mass Index (BMI) z-scores, which measure the relative weight that is adjusted for the age and gender of the child [20]. Accordingly, a child with a BMI z-score > 1, BMI z-score > 2 and BMI z-score > 3 is classified respectively as a risk to be overweight child, an overweight child and an obese child. The primary explanatory variable is mother smoking frequency (*M_smoking*), measured as the number of cigarettes smoked by a mother daily. The NIDS records each individual's height, weight, and waist measurement in the database, allowing us to construct the BMI using the weight (in kg) and the height in meters squared ($m^2$). The study controls for the mothers' weight categorised into average weight, overweight, obese, and hyper obese, following the World Health Organisation index [20]. A person is classified as overweight if their BMI lies between 25 and 30, obese if BMI > 30, and hyper-obese if BMI > 35.

South Africa is a culturally diverse country comprising four racial categories (African, White, Coloured, and Asian/Indian) with huge socioeconomic differences. Because of the high heterogeneity between South Africans and consistent with previous studies, the analysis controls for cultural, sociodemographic and economic characteristics of households. These include child grant recipient (*C_grant*), the gender of the child (*Boy*), child birthweight (*C_birthweight*), age of the mother (*M_age*), geographical areas (*Urban*), marital status of the mother (*M_maritalstatus*), household size (*Hhsize*), father overweight (*F_overweight*), mother's weight (*M_overweight*, *M_obese*, *M_hyperobese*, *Average*), and race (*White*, *Asian/Indian*, *Coloured* and *Black*). The details on variables definitions and summary statistics are provided in Table 1.

## 2.2 Empirical strategy

The association between maternal smoking and the prevalence of childhood overweight is examined using an inferential analysis from the panel logistic regression model of the following specification:

$$Y_{it} = \alpha_0 + \alpha_1 M\_smoking_{it} + \beta X_{it} + a_i + e_{it} \tag{1}$$

Where $Y_{it}$ is the health variable of child $i$ in wave $t$ proxied by: the risk of child overweight, child overweight, or child obesity. $M\_smoking_{it}$ is mother's smoking frequency, $X_{it}$ is the vector of control variables, $a_i$ is the unobserved child characteristics also known as unobserved heterogeneity, and $e_{it}$ is the idiosyncratic term.

Because $Y_{it}$ is a binary outcome variable, Eq (1) aims at estimating the probability of the realization of child weight abnormalities, conditioned in terms of the maternal smoking status

**Table 1. Descriptive statistics of variables used in the estimation.**

| Variables | Description of variables | Mean/Frequency | Std. Dev. |
|---|---|---|---|
| *Dependent Variables* | | | |
| **Child Health** | | | |
| Risk of Child Overweight | BMI-for-age Z-score > 1.0 | .82 | .37 |
| Child Overweight | BMI-for-age Z-score >2.0 | .76 | .42 |
| Child Obese | BMI-for-age Z-score >3.0 | .74 | .43 |
| *Explanatory variable* | | | |
| **Smoking Variable** | | | |
| Maternal Smoking | Number of cigarettes smoked per day | 7.70 | 6.44 |
| **Maternal characteristics** | | | |
| Maternal age | Age of the mother (continuous variable) | 42.36 | 14.14 |
| Married mother | Married mother (Yes = 1) | 0.44 | 0.49 |
| Maternal overweight | BMI_mother > = 25 | .90 | .28 |
| Maternal obese | BMI_mother > = 30 | .82 | .37 |
| Maternal hyperobese | BMI_mother > = 35 | .75 | .43 |
| **Covariates of outcome model** | | | |
| Child gender | Boy (Yes = 1) | .50 | .50 |
| Urban | Urban (Yes = 1) | .48 | .49 |
| Child birth weight | Birth weight of child (continuous) | 3.33 | 9.58 |
| African | Race category 1 (Yes = 1) | .79 | .40 |
| Coloured | Race category 2 (Yes = 1) | .13 | .34 |
| Indian | Race category 3 (Yes = 1) | .01 | .12 |
| White | Race category 4 (Yes = 1) | .04 | .21 |
| Father overweight | BMI_mother > = 25 | .97 | .15 |

and the rest of the covariates. To this end, a logistic function is used to constrain the estimated probability of lying between 0 and 1, as it is standard in the literature. As a logistic probability model, Eq 1 can be rewritten as follows:

$$P(Y_{it} = 1 | M\_smoking_{it}, X_{it}) = \Lambda(\alpha_0 + \alpha_1 M\_smoking_{it} + \beta X_{it} + a_i + e_{it}) \quad (2)$$

The cumulative distribution function of the standard logistic distribution, $\Lambda(z) = \frac{1}{1+exp(-z)}$, lies between 0 and 1; therefore, the logit estimates also lie between 0 and 1. The logit regression is estimated using the maximum likelihood estimators as the parameters $a_k$, $k = 0, 1$, and $\beta_j$, $j = 1, \ldots, k$ is nonlinear. Therefore, strict linearity between child weight and smoking frequency is not required. Also, normality and homoscedasticity are not compulsory standards for this model. The logistic regression requires the non-existence of multicollinearity among the independent variables, and a direct relationship between the log odds and the independent variables.

The panel framework offers two different estimators for Eq 2 that account for unobserved heterogeneity: the fixed effects (FE) and the random effects (RE) estimators. However, the FE estimator does accommodate time-invariant covariates due to demeaning transformation, allowing the difference in the unobserved heterogeneity. Thus, we use a random effect estimator to accommodate significant covariates included in Eq 2, namely gender, race, and geographical areas that are all time-invariant.

### 2.3 Ethical statement

Ethics Approval for data collection for the five waves was granted by the University of Cape Town's Commerce Faculty Ethics in Research Committee on 12 December 2007 with additional approval for wave 5 granted by the Faculty of Health Sciences Human Research Ethics Committee on 19 December 2016. All the datasets are available in the public domain, and all information about respondent identities, used to track participants over time, has been excluded from the public release data.

For all the waves, the survey paper consent forms are issued in all official languages during the interviews and the informed consent process is conducted in the respondent's language of choice. The consent forms for children and young adults (15–17 years old) are signed by their caregivers. This was further supplemented in wave 5 by the verbal consent of the proxy respondents to allow the interview to be conducted on their behalf. In addition, the assent forms for children anthropometric measures require children aged between 7 to 10 years old to indicate their willingness to be measured. The interviews with missing signed consent forms are excluded from the dataset [21].

## 3. Results

### 3.1. Descriptive analysis

Descriptive statistics show that approximately 82 percent children in the sample are at risk of being overweight, 76 percent are overweight and 74 percent are obese across the sample period. While the sampled children are gender balanced, the racial representation is 79 percent African, 13 percent Coloured, 1 percent Indian, and 4 percent White. In terms of maternal characteristics, approximately 48 percent of mothers live in urban areas, 90 percent of whom are overweight, 82 percent are obese, and 75 percent are hyper-obese. Mothers smoke on average a minimum of seven cigarettes daily and about 44 percent of them are married. In terms of the hypothesized correlation, Fig 1 depicts similar trends in child nutritional health measures

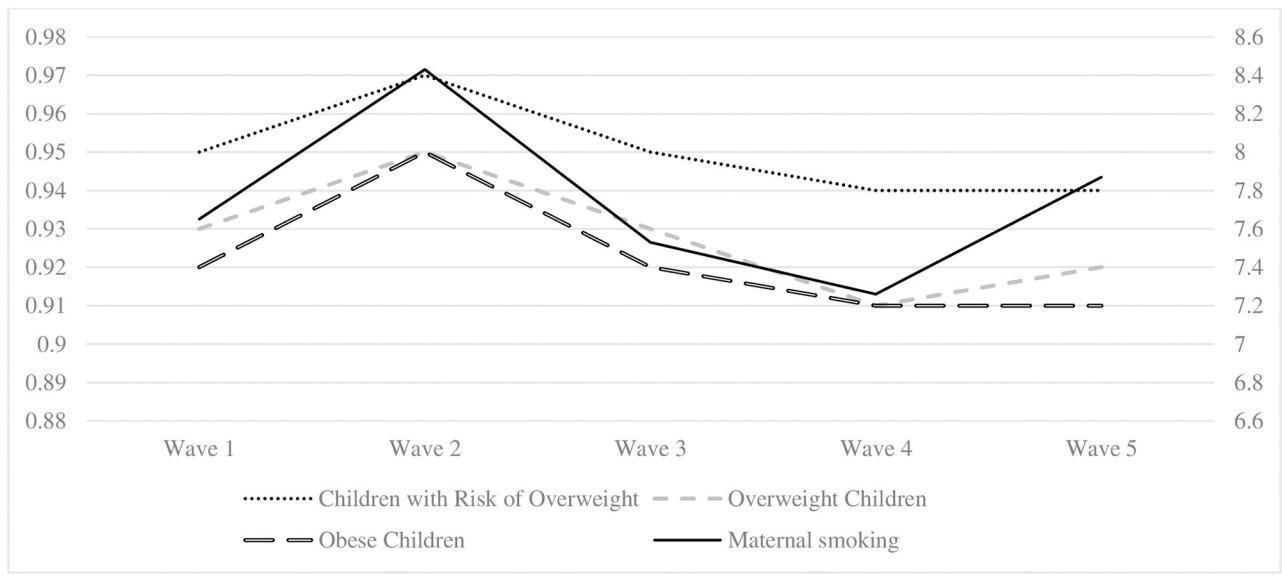

**Fig 1. Trend in childhood overweight prevalence and maternal smoking across waves.** Note. On the left primary axis are the three child health variables measured by the average percentages of children with risk of overweight, overweight children and obese children, respectively. Maternal smoking frequency is depicted on the secondary axis on the right, measuring the average daily number of cigarettes smoked by mothers.

and mothers' smoking frequency across time. Consistently with maternal smoking, risk of obesity and obesity in children appear to have experienced a similar jump between wave 1 and wave 2, followed by a decreasing trend from wave 2 to wave 4. In the most recent wave, these dynamics have reversed with the rise of maternal smoking frequency as well as the increase fraction of overweight children although the percentage of children with the risk of overweight and the proportion of obese children have remained constant.

## 3.2. Maternal sociodemographic determinants of child weight abnormalities

The estimates of the effect of maternal smoking on each of the three child overweight prevalence proxies are summarized in Tables 2 and 3 where the dependent variable is respectively, risk of child overweight (Model 1), child overweight (Model 2) and child obese (Model 3). Each of the models regresses a child overweight prevalence proxy on maternal smoking, adjusting for control variables (child grant, the gender of the child, child birthweight, age of the mother, geographical areas, marital status of the mother, household size, father overweight, mother's weight categories and different race groups).

Table 2 displays the estimated odds ratio with random effects, which allow to control for unobserved heterogeneity while accommodating for time invariant variables such as race. Across the three models, maternal smoking is significantly associated with an increased risk of child overweight (odds ratio 0.02), child overweight (odds ratio 0.02) and child obese (odds ratio 0.02). Expectedly, the results also reveal that child nutritional health significantly exhibits positive odds with father overweight and mother obesity but negative odds with mother overweight. However, because the odds ratio bears less meaning with regard to the estimated probability, marginal effects displayed in Table 3 are provided for inference. Consistently with the odds ratio, maternal smoking has a positive association with child weight abnormalities. One extra cigarette smoked daily by a mother raises the odds of the risk of overweight, overweight and obesity in children by 0.0031, 0.0036 and 0.0042 basis points, respectively (Model 1, Model 2 and Model 3). These findings confirm the proposition that parental life style plays an important role in explaining offspring health at their early stage of life. Besides maternal smoking, mother' characteristics, including age, marital status, and weight abnormalities, display a consistent significant association with child having high BMIs. Mother's age and maternal hyper obesity status increase the probability of weight irregularities in children while being married has a mitigating effect on the likelihood of obesity, overweight and risk of overweight in children. Unsurprisingly, household size decreases the probability of weight abnormalities of children given the competing nutritional needs of children associated with a reduced risk of overfeeding as household size increases. Similarly, children with low birth weight rate are less likely to experience odd weight. In terms of population group, race does indeed have an impact on the odds of child weight issues, but only on the risk of overweight in children (Model 1), not on overweight (Model 2), or obesity (Model 3). In particular, being Asian/Indian reduces the likelihood of the child overweight risk, which might be attributed to their relatively healthy eating behaviour. Important to note also is the fact that living in urban areas and receiving a child grant are associated with a decreased odds and/or probability of weight abnormalities in children, yet with a generally weak association and/or inconsistently across the model.

## 4. Discussion

The empirical findings reveal that maternal sociodemographic characteristics play a significant role in explaining child weight abnormalities in South Africa. Particularly, a positive association is found between maternal smoking and nutritional child health outcome, regardless of

**Table 2. Odds of maternal smoking on childhood overweight prevalence.**

| | Model 1 | Model 2 | Model 3 |
|---|---|---|---|
| | Child risk of overweight | Child overweight | Child obese |
| Maternal smoking | 0.02** | 0.02** | 0.02*** |
| | (0.01) | (0.01) | (0.01) |
| Child_birthweight | -0.00** | -0.00*** | -0.00*** |
| | (0.00) | (0.00) | (0.00) |
| Maternal age | 0.07 | -0.02 | 0.02 |
| | (0.11) | (0.11) | (0.11) |
| Married mother | 0.09*** | 0.11*** | 0.12*** |
| | (0.00) | (0.01) | (0.01) |
| Maternal overweight | -0.35** | -0.26* | -0.31** |
| | (0.14) | (0.13) | (0.13) |
| Maternal obese | 0.34** | 0.19 | 0.16 |
| | (0.15) | (0.14) | (0.14) |
| Maternal hyperobese | -0.07 | 0.01 | -0.02' |
| | (0.18) | (0.17) | (0.17) |
| Father overweight | 0.46** | 0.43** | 0.35* |
| | (0.20) | (0.19) | (0.19) |
| Household size | 0.10 | 0.17 | 0.20 |
| | (0.18) | (0.18) | (0.18) |
| Urban | -0.04** | -0.04** | -0.03* |
| | (0.02) | (0.02) | (0.02) |
| Child_grant | -0.18 | -0.25 | -0.26* |
| | (0.16) | (0.15) | (0.16) |
| Boy | -0.31** | -0.24* | -0.15 |
| | (0.13) | (0.13) | (0.13) |
| Coloured | -0.14 | 0.08 | 0.16 |
| | (0.15) | (0.14) | (0.15) |
| Asian/Indian | -1.15*** | -0.59 | -0.38 |
| | (0.42) | (0.42) | (0.43) |
| White | -0.23 | 0.00 | 0.00 |
| | (0.26) | (0.25) | (0.25) |
| Constant | -1.14*** | -2.26*** | -3.07*** |
| | (0.39) | (0.40) | (0.43) |
| LR test (p-value) | 0.00 | 0.00 | 0.00 |
| log likelihood ratio | -1224.98 | -1333.23 | -1361.5 |
| Observation | 2,484 | 2,484 | 2,484 |

*Figures displayed are odds of child health on maternal smoking and other covariates based on random effects. Models 1, 2, and 3 use the risk of overweight, overweight, and obese as a proxy for child health, respectively. †***, **, * denote statistical significance at a 1%, 5% and 10% level of significance, respectively. ‡Standard errors are reported in parentheses

the proxy used. These results are in line with previous findings that children of smoking mothers are more likely to be overweight or obese than those of smoke-free mothers [10, 22]. This is understandable since smoking may lead to depression in mothers with overreaching consequences on child health either directly through unhealthy feeding scheme and nutritional choices or indirectly through genetic transfer. The genetic channel implies a transfer of parental weight abnormalities to their offspring; translating into a positive correlation between

**Table 3. Marginal effects of maternal smoking on childhood overweight prevalence.**

| | Model 1 | Model 2 | Model 3 |
|---|---|---|---|
| | Child risk of overweight | Child overweight | Child obese |
| Maternal smoking | 0.00** | 0.00** | 0.00*** |
| | (0.00) | (0.00) | (0.00) |
| Child_birth weight | -0.00 | -0.00*** | -0.00*** |
| | (0.00) | (0.00) | (0.00) |
| Maternal age | 0.01*** | 0.01*** | 0.02*** |
| | (0.00) | (0.00) | (0.00) |
| Married mother | -0.05** | -0.04* | -0.04** |
| | (0.02) | (0.02) | (0.02) |
| Maternal overweight | 0.04** | 0.03 | 0.02 |
| | (0.02) | (0.02) | (0.02) |
| Maternal obese | -0.01 | 0.00 | -0.00 |
| | (0.02) | (0.02) | (0.02) |
| Maternal hyperobese | 0.06** | 0.06** | 0.05* |
| | (0.02) | (0.03) | (0.02) |
| Father overweight | 0.01 | 0.02 | 0.03 |
| | (0.02) | (0.02) | (0.02) |
| Household size | 0.00* | -0.00** | -0.00* |
| | (-0.00) | (0.00) | (0.00) |
| Urban | 0.02 | -0.03 | -0.04* |
| | (0.02) | (0.02) | (0.02) |
| Child_grant | -0.04** | -0.03* | -0.02 |
| | (0.01) | (0.02) | (0.02) |
| Boy | 0.00 | -0.00 | 0.00 |
| | (0.00) | (0.01) | (0.01) |
| Coloured | -0.02 | 0.01 | 0.02 |
| | (0.02) | (0.02) | (0.00) |
| Asian/Indian | -0.18*** | -0.10 | -0.06 |
| | (0.07) | (0.07) | (0.07) |
| White | -0.03 | 0.00 | 0.00 |
| | (0.03) | (0.03) | (0.04) |

*The figures displayed are marginal effects from the random effects logit regression model of child health on maternal smoking and other covariates. Models 1, 2, and 3 use the risk of overweight, overweight, and obesity as a proxy for child health, respectively. †***, **, * denote statistical significance at a 1%, 5% and 10% level of significance, respectively. ‡Standard errors are reported in parentheses.

children weight and that of their parents. The results substantiate this assumption as mother's hyper-obese status appears to be a significant driver of weight anomalies in children. Therefore, regular smoking mothers are at a greater risk of contributing to non-communicable diseases in their offspring. These non-communicable diseases include overweight or obesity, that is transferable from smoking parents, or being born from parents who suffer from overweight.

In addition, mothers' age, marital status and child grant recipient, display a consistent significant association with child health [23, 24]. It is found that weight irregularities in children tend to increase with mothers' age irrespective of the specification. This could probably indicate the inability to remain and/or follow a healthy life style as age increases with spillover effect on child care. It is also possible that age related weights gain in mothers can be transmitted to their children, namely those born by older mothers. Conversely, it is shown that children

from married mothers are less exposed to weight problems compared to those from their unmarried counterparts. This finding could be attributed to the health benefit of a united family, including, reduced stress and anxiety, increased happiness and healthy life style, which boost the physical and emotional well-being [25]. Finally, child grant appears to reduce the likelihood of child weight problems. This finding suggests that child support programs can be a life-saving tool for children coming from poor households by releasing financial pressure to some extent, with positive influence on nutrition, which further improves child development. Notwithstanding the fact that living in urban areas expectedly offers better exposure to education and an awareness of malnutrition, with a possible reduction in the probability of being overweight and providing better care to offspring, "Urban" variable depicts a marginal decrease in the probability of children to be obese. Similarly, being Asian/Indian is significantly associated with the negative probability of the risk of being overweight in children with not on child overweight and obesity. This suggests a rather marginal racial difference in the prevalence of childhood overweight in South Africa.

However, the above inference is subject to the issue of attrition prevalent in dynamic surveys like NIDS dataset, which is attributed to the respondents' relocation, death or non-responses. Moreover, because of data restrictions, the empirical strategy could not control for diet, which is claimed to have a significant influence on health challenges in children. The use of alternative data sources as they become available, could help mitigate these issues while providing further insight on the prevalence of childhood obesity in South Africa.

## 5. Conclusion

This study assesses the link between maternal smoking frequency (number of cigarettes smoked per day) and childhood weight status in South Africa, using a random-effect logic model implemented on a panel dataset made of the five NIDS waves. Overall, evidence shows that maternal smoking contributes to child overweight and/or obesity. In addition, the mother's hyperobese status is a significant driver of weight anomalies in children, which motivates the importance of early interventions targeted at mothers. The results further indicate the influence of childhood issues and exposure to harsh early life factors on a child's weight status.

Because of the complexity of overweight challenges, there is a need to strengthen population health policies linked to implementing educational programs to minimise tobacco consumption. Policymakers can prioritise programs to reduce smoking behaviour, especially amongst pregnant women and caregivers, to minimise the risk of childhood overweight. Furthermore, comprehensive action plans and recommendations that address norms and make treatment plans available for the children who are already suffering from overweight, are essential. These include child support programs, the promotion and the consumption of healthy foods accompanied by physical activity, which may reduce mothers' stress levels and their incentive to self-medicate using tobacco substances. Finally, Parents must be made aware of the different adverse outcomes of tobacco smoking as one of the most preventable causes of childhood overweight or obesity, among other child-related development complications.

## Acknowledgments

The authors express their gratitude to the four anonymous reviewers and the editor for their insightful comments, which have helped improve the quality of their paper.

## Author Contributions

**Conceptualization:** Nomusa Y. Nkomo, Beatrice D. Simo-Kengne, Mduduzi Biyase.

**Data curation:** Nomusa Y. Nkomo.

**Formal analysis:** Nomusa Y. Nkomo, Beatrice D. Simo-Kengne, Mduduzi Biyase.

**Investigation:** Nomusa Y. Nkomo.

**Methodology:** Nomusa Y. Nkomo, Beatrice D. Simo-Kengne, Mduduzi Biyase.

**Supervision:** Beatrice D. Simo-Kengne, Mduduzi Biyase.

**Validation:** Beatrice D. Simo-Kengne, Mduduzi Biyase.

**Writing – original draft:** Nomusa Y. Nkomo.

**Writing – review & editing:** Nomusa Y. Nkomo, Beatrice D. Simo-Kengne, Mduduzi Biyase.

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
