## [Decision Letter · Decision Letter 0]

23 May 2022

PONE-D-22-12309Impact of maternal tobacco smoking on childhood obesity in South AfricaPLOS ONE

Dear Dr. Simo-Kengne,

Thank you for submitting your manuscript to PLOS ONE. After careful consideration, we feel that it has merit but does not fully meet PLOS ONE’s publication criteria as it currently stands. Therefore, we invite you to submit a revised version of the manuscript that addresses the points raised during the review process.

ACADEMIC EDITOR:The topic of the research is important. However it's necessary for this research paper to has accurate scientific revision  and modifications from the authors specially in the method section which needs scientific reformulation and organization under subtitles, also the results section it  needs necessary special attention from the authors  " Table 1 particularly" as well as structure of the discussion must be conducted.The reviewers comments are very helpful to the authors to improve their research paper to be suitable for publication.

We look forward to receiving your revised manuscript.

Kind regards,

Omnia Samir El Seifi, Professor

Academic Editor

PLOS ONE

Journal Requirements:

3.Thank you for stating the following financial disclosure: 

Additional Editor Comments :

The topic of the research is important.  The method section needs scientific reformulation and organization under subtitles, please give the results special attention " Table 1 particularly" as well as structure of the discussion must be conducted.

Reviewers' comments:

Reviewer's Responses to Questions

**Comments to the Author**

1. Is the manuscript technically sound, and do the data support the conclusions?

Reviewer #1: Yes

Reviewer #2: No

Reviewer #3: Yes

Reviewer #4: Yes

2. Has the statistical analysis been performed appropriately and rigorously? 

Reviewer #1: Yes

Reviewer #2: No

Reviewer #3: Yes

Reviewer #4: Yes

3. Have the authors made all data underlying the findings in their manuscript fully available?

Reviewer #1: Yes

Reviewer #2: No

Reviewer #3: No

Reviewer #4: Yes

4. Is the manuscript presented in an intelligible fashion and written in standard English?

Reviewer #1: Yes

Reviewer #2: No

Reviewer #3: No

Reviewer #4: No

5. Review Comments to the Author

Reviewer #1: - Title - Impact of maternal tobacco smoking on childhood obesity in South Africa - suggestion: Maternal tobacco smoking and childhood obesity: a cohort study

abstract:

- review purpose - this study examines the affiliation... I would suggest to use the word association.

- review Methods: "The link between maternal tobacco consumption and child health was assessed using logistic regression models in which unobserved child characteristics are assumed to be uncorrelated with independent variables. This random effect assumption allows us to control for time-invariant sociodemographic factors in estimating the probability of overweight prevalence in children. " - it does not describe the study design, variables studied, sample size calculation, inclusion and exclusion criteria.

- The results suggest... this is not appropriate to use the term suggest.

- Introduction is fine, although the last sentence that one would expect the objetive: "Therefore, understanding the contributing factors to childhood obesity becomes imperative in addressing overweight-related future public health challenges in South Africa." - it is not very straight forward as an objective - it should be reviewed.

- There are some essential aspects that should be covered in the methods section.

- The description of the result section are very superficial and do not cover the most relevant results

- The first sentence of the discussion is not well presented as you would expect that this sentence should be the main result of the study: "Because the odd ratio bears less meaning with regard to the estimated probability, marginal effects displayed in Table 3 are used for inference."

- This sentence is not clearly presented: "The outcome of the study shows that, as the frequency of

maternal smoking increases, the likelihood for children to have abnormal weight also increases." - it has to be reviewed.

- The discussion is very superficial and needs to compare the results of the present study with the literature.

- No limitation of the study has been presented.

Reviewer #2: Dear authors’ thank you for submitting this nice manuscript.

Here under some comments about your manuscript.

1. Try to summarize and minimize your manuscript introduction part.

2. Write your study design clearly.

3. Your dependent and independent variables are not clearly stated. List them.

4. What is the meaning of Table 1??? It is meaningless, because it is known that computing the mean, sd, minimum and maximum values for categorical variables are impossible. So remove this table, unless revise it.

5. Your results in the table 2 and table 3 are not correctly presented. I think that it is meaningless. Where is the estimates of the parameters? Where is the interpretation of the parameters?

6. General try to re-write your manuscript.

Reviewer #3: Dear author, I attached some of my comments here.

-_You have to put line number for the whole part of your manuscript

Abstract

-You did not included introduction.

-The method part does not include important information like Sample size, sampling technique, study design, the major assumptions checked and the measures of association used.

Methods

-Why you put all portions of the method in one paragraph. It is better to put in different sub-headings.

-Although it is secondary data, for the reader of the paper to be clear you have to elaborate about the sampling method, the way you reach the sample size, data collection tools and procedures.

-What about the inclusion and exclusion criteria, data quality control and ethical assurance?

-You didn’t described about the models (what is included in model1, model2 and model3 and what criteria was used to compare those models and which one is better)

-What measure of association you had used?

Result

-Why did you prefer using mean and standard deviation?

-The variables did not have category

-The tables need full title, which describes about place, person, time and the problem. In addition, above the table you have to describe major things from the table using texts.

- In result section, Interpretation of the finding is mandatory

Discussion

-you did not included the possible justifications

Reviewer #4: The authors conducted a large-scale study examining the association between maternal tobacco use and children's health in South Africa in terms of body weight condition. In general terms the article is clearly stated, well documented, easy to follow.

However, there are some minor issues that the authors could address:

-The introduction presents the scientific background very explanatory, but I think it is necessary that the authors state more clearly in this section the specific objectives and hypotheses.

-I suggest that to reduce the ambiguity of this paper, the authors should present the methods in a structured way, in line with the STROBE Statement for observational studies. Also, in the methods section the authors should present the study design, settings, and participant eligibility criteria. These seem to be missing or are presented ambiguously.

- In the results section, the descriptive statistics presented in Table 1 appear to be incorrect. The authors should present categorical variables as numbers (percentage, confidence interval).

-Regression models should also be explained in more detail, clearly showing unstandardized and standardized regression weights and p-values. Odds ratios could also be presented.

-The discussion summarizes the results, but I think the conclusions are very long. Much of that presented in the conclusions section should be moved to the discussion.

6. PLOS authors have the option to publish the peer review history of their article (what does this mean?). If published, this will include your full peer review and any attached files.

Reviewer #1: **Yes: **no

Reviewer #2: No

Reviewer #3: No

Reviewer #4: No

---

## [Author Response · Author response to Decision Letter 0]

12 Jul 2022

Dear anonymous reviewers, thank you for your insightful comments and suggestions which have helped improve the quality of our manuscript. The revision report is attached, which provides detailed feedback to each of your valuable comments. Your contribution is gratefully acknowledged.

---

## [Decision Letter · Decision Letter 1]

18 Oct 2022

Maternal tobacco smoking and childhood obesity in South Africa: A cohort study

PONE-D-22-12309R1

Dear Dr. Simo-Kengne,

We’re pleased to inform you that your manuscript has been judged scientifically suitable for publication and will be formally accepted for publication once it meets all outstanding technical requirements.

Kind regards,

Omnia Samir El Seifi, Professor

Academic Editor

PLOS ONE

Additional Editor Comments (optional):

Reviewers' comments:

Reviewer's Responses to Questions

**Comments to the Author**

1. If the authors have adequately addressed your comments raised in a previous round of review and you feel that this manuscript is now acceptable for publication, you may indicate that here to bypass the “Comments to the Author” section, enter your conflict of interest statement in the “Confidential to Editor” section, and submit your "Accept" recommendation.

Reviewer #5: All comments have been addressed

2. Is the manuscript technically sound, and do the data support the conclusions?

Reviewer #5: Yes

3. Has the statistical analysis been performed appropriately and rigorously? 

Reviewer #5: I Don't Know

4. Have the authors made all data underlying the findings in their manuscript fully available?

Reviewer #5: Yes

5. Is the manuscript presented in an intelligible fashion and written in standard English?

Reviewer #5: Yes

6. Review Comments to the Author

Reviewer #5: The authors have revised the manuscript successfully, so I recommend this paper for possible publication.

7. PLOS authors have the option to publish the peer review history of their article (what does this mean?). If published, this will include your full peer review and any attached files.

Reviewer #5: No

---

## [Editor Report · Acceptance letter]

20 Oct 2022

PONE-D-22-12309R1 

Maternal tobacco smoking and childhood obesity in South Africa: A cohort study 

Dear Dr. Simo-Kengne:

I'm pleased to inform you that your manuscript has been deemed suitable for publication in PLOS ONE. Congratulations! Your manuscript is now with our production department. 

Kind regards, 

on behalf of

Professor Omnia Samir El Seifi 

Academic Editor

PLOS ONE